# Minority-centric meta-analyses of blood lipid levels identify novel loci in the Population Architecture using Genomics and Epidemiology (PAGE) study

Yao Hu[1‡], Mariaelisa Graff[2‡], Jeffrey Haessler[1], Steven Buyske[3], Stephanie A. Bien[1], Ran Tao[4,5], Heather M. Highland[2], Katherine K. Nishimura[1], Niha Zubair[1], Yingchang Lu[6], Marie Verbanck[6], Austin T. Hilliard[7], Derek Klarin[8,9,10], Scott M. Damrauer[11,12,13], Yuk-Lam Ho[14], the VA Million Veteran Program[¶], Peter W. F. Wilson[11,15], Kyong-Mi Chang[12,16], Philip S. Tsao[17,18], Kelly Cho[14], Christopher J. O'Donnell[14,19], Themistocles L. Assimes[17,18], Lauren E. Petty[5,20], Jennifer E. Below[5,20], Ozan Dikilitas[21], Daniel J. Schaid[22], Matthew L. Kosel[22], Iftikhar J. Kullo[21], Laura J. Rasmussen-Torvik[23], Gail P. Jarvik[24], Qiping Feng[25], Wei-Qi Wei[25], Eric B. Larson[26], Frank D. Mentch[27], Berta Almoguera[27], Patrick M. Sleiman[27], Laura M. Raffield[28], Adolfo Correa[29], Lisa W. Martin[30], Martha Daviglus[31,32], Tara C. Matise[3], Jose Luis Ambite[33], Christopher S. Carlson[1], Ron Do[6], Ruth J. F. Loos[6], Lynne R. Wilkens[34], Loic Le Marchand[34], Chris Haiman[35], Daniel O. Stram[35], Lucia A. Hindorff[36], Kari E. North[2], Charles Kooperberg[1], Iona Cheng[37]*, Ulrike Peters[1]*

1 Public Health Sciences Division, Fred Hutchinson Cancer Research Center, Seattle, Washington, United States of America, 2 Department of Epidemiology, University of North Carolina at Chapel Hill, Chapel Hill, North Carolina, United States of America, 3 Department of Statistics and Biostatistics, Rutgers University, New Brunswick, New Jersey, United States of America, 4 Department of Biostatistics, Vanderbilt University Medical Center, Nashville, Tennessee, United States of America, 5 The Vanderbilt Genetics Institute, Division of Genetic Medicine, Vanderbilt University Medical Center, Nashville, Tennessee, United States of America, 6 The Charles Bronfman Institute for Personalized Medicine, The Icahn School of Medicine at Mount Sinai, New York, New York, United States of America, 7 Palo Alto Veterans Institute for Research, VA Palo Alto Health Care System, Palo Alto, California, United States of America, 8 Center for Genomic Medicine, Massachusetts General Hospital, Harvard Medical School, Boston, Massachusetts, United States of America, 9 Program in Medical and Population Genetics, Broad Institute of MIT and Harvard, Cambridge, Massachusetts, United States of America, 10 Boston VA Healthcare System, Boston, Massachusetts, United States of America, 11 Emory Clinical Cardiovascular Research Institute, Atlanta, Georgia, United States of America, 12 Corporal Michael Crescenz VA Medical Center, Philadelphia, Pennsylvania, United States of America, 13 Department of Surgery, Perelman School of Medicine, University of Pennsylvania, Philadelphia, Pennsylvania, United States of America, 14 Massachusetts Veterans Epidemiology Research and Information Center (MAVERIC), VA Boston Healthcare System, Boston, Massachusetts, United States of America, 15 Atlanta VA Medical Center, Decatur, Georgia, United States of America, 16 Department of Medicine, Perelman School of Medicine, University of Pennsylvania, Philadelphia, Pennsylvania, United States of America, 17 Department of Medicine, Stanford University School of Medicine, Stanford, California, United States of America, 18 VA Palo Alto Health Care System, Palo Alto, California, United States of America, 19 Department of Medicine, Harvard Medical School, Boston, Massachusetts, United States of America, 20 Department of Epidemiology, Human Genetics & Environmental Sciences, University of Texas School of Public Health, Houston, Texas, United States of America, 21 Department of Cardiovascular Medicine, Mayo Clinic, Rochester, Minnesota, United States of America, 22 Department of Health Sciences Research, Mayo Clinic, Rochester, Minnesota, United States of America, 23 Department of Preventive Medicine, Northwestern University Feinberg School of Medicine, Chicago, Illinois, United States of America, 24 Department of Medicine, University of Washington Medical Center, Seattle, Washington, United States of America, 25 Department of Medicine, Division of Clinical Pharmacology, Vanderbilt University Medical Center, Nashville, Tennessee, United States of America, 26 Kaiser Permanente Washington Health Research Institute, Seattle, Washington, United States of America, 27 Center for Applied Genomics, Children's Hospital of Philadelphia, Philadelphia, Pennsylvania, United States of America, 28 Department of Genetics, University of North Carolina at Chapel Hill, Chapel Hill, North Carolina, United States of America, 29 Departments of Medicine, Pediatrics, and Population Health Science, University of Mississippi Medical



**Data Availability Statement:** Individual level phenotype and genotype data are available through dbGAP (phs000356.v2.p1).

**Funding:** The PAGE program is funded by the National Human Genome Research Institute (NHGRI) with co-funding from the National Institute on Minority Health and Health Disparities (NIMHD). The contents of this paper are solely the responsibility of the authors and do not necessarily represent the official views of the NIH. The PAGE consortium thanks the staff and participants of all PAGE studies for their important contributions. We thank Rasheeda Williams and Margaret Ginoza for providing assistance with program coordination. The complete list of PAGE members can be found at http://www.pagestudy.org. Assistance with data management, data integration, data dissemination, genotype imputation, ancestry deconvolution, population genetics, analysis pipelines, and general study coordination was provided by the PAGE Coordinating Center (NIH U01HG007419). Genotyping services were provided by the Center for Inherited Disease Research (CIDR). CIDR is fully funded through a federal contract from the National Institutes of Health to The Johns Hopkins University, contract number HHSN268201200008I. Genotype data quality control and quality assurance services were provided by the Genetic Analysis Center in the Biostatistics Department of the University of Washington, through support provided by the CIDR contract. The current study included participants from BioMe, HCHS/SOL, MEC and WHI. BioMe: Data of BioMe Biobank used in this study was provided by the Charles Bronfman Institute for Personalized Medicine at the Icahn School of Medicine at Mount Sinai. Phenotype data collection was supported by the Andrea and Charles Bronfman Philanthropies. Funding support for the PAGE BioMe study was provided through the National Human Genome Research Institute (NIH U01HG007417). HCHS/SOL: Primary funding support to Dr. North and colleagues is provided by U01HG007416. Additional support was provided via R01DK101855 and 15GRNT25880008. The HCHS/SOL study was carried out as a collaborative study supported by contracts from the National Heart, Lung, and Blood Institute (NHLBI) to the University of North Carolina (N01-HC65233), University of Miami (N01-HC65234), Albert Einstein College of Medicine (N01-HC65235), Northwestern University (N01-HC65236), and San Diego State University (N01-HC65237). The following Institutes/Centers/Offices contribute to the HCHS/SOL through a transfer of funds to the NHLBI: NIMHD, National Institute on Deafness and Other Communication Disorders, National Institute of Dental and Craniofacial Research, National Institute of Diabetes and Digestive and Kidney Diseases, National Institute of Neurological Disorders and Stroke, NIH Institution-Office of

Center, Jackson, Mississippi, United States of America, **30** School of Medicine and Health Sciences, George Washington University, Washington, District of Columbia, United States of America, **31** Institute for Minority Health Research, University of Illinois at Chicago, Chicago, Illinois, United States of America, **32** Department of Medicine, University of Illinois at Chicago, Chicago, Illinois, United States of America, **33** Information Sciences Institute, University of Southern California, Marina del Rey, California, United States of America, **34** Epidemiology Program, University of Hawaii Cancer Center, Honolulu, Hawaii, United States of America, **35** Keck School of Medicine, University of Southern California, Los Angeles, California, United States of America, **36** Division of Genomic Medicine, NIH National Human Genome Research Institute, Bethesda, Maryland, United States of America, **37** Cancer Prevention Institute of California, Fremont, California, United States of America

‡ These authors share first authorship on this work.
¶ Membership of the VA Million Veteran Program is available in S1 Text.
* ICheng@psg.ucsf.edu (IC); upeters@fredhutch.org (UP)

## Abstract

Lipid levels are important markers for the development of cardio-metabolic diseases. Although hundreds of associated loci have been identified through genetic association studies, the contribution of genetic factors to variation in lipids is not fully understood, particularly in U.S. minority groups. We performed genome-wide association analyses for four lipid traits in over 45,000 ancestrally diverse participants from the Population Architecture using Genomics and Epidemiology (PAGE) Study, followed by a meta-analysis with several European ancestry studies. We identified nine novel lipid loci, five of which showed evidence of replication in independent studies. Furthermore, we discovered one novel gene in a PrediXcan analysis, minority-specific independent signals at eight previously reported loci, and potential functional variants at two known loci through fine-mapping. Systematic examination of known lipid loci revealed smaller effect estimates in African American and Hispanic ancestry populations than those in Europeans, and better performance of polygenic risk scores based on minority-specific effect estimates. Our findings provide new insight into the genetic architecture of lipid traits and highlight the importance of conducting genetic studies in diverse populations in the era of precision medicine.

## Author summary

Blood lipid levels are closely linked to cardio-metabolic diseases, and genetic factors play an important role in their metabolism and regulation. Although over 400 loci have been identified through genetic association studies, the genetic architecture of lipid levels is not fully characterized. The lack of representation of diverse populations in previous studies resulted in a large gap in understanding the genetic background of lipid traits between European and minority populations, including African Americans, Hispanics, Hawaiians, and Native Americans. In our current analyses which included ancestrally diverse populations, we identified nine novel loci, one novel gene, and minority-specific independent signals at eight known loci, and pinpointed potential functional variants at two known loci. We further observed smaller effect sizes of reported lipids-associated loci in African Americans and Hispanics than those in Europeans, and better performance of polygenic risk scores using minority-specific instead of European-derived effect sizes when estimating genetic predisposition in minority populations. Our findings showed the benefits of

Dietary Supplements. MEC: The epidemiological architecture of the MEC study is funded through the NHGRI PAGE program (NIH U01HG007397). The MEC study is funded through the National Cancer Institute U01CA164973. WHI: Funding support for the "Exonic variants and their relation to complex traits in minorities of the WHI" study is provided through the NHGRI PAGE program (NIH U01HG007376). The WHI program is funded by the NHLBI, NIH, U.S. Department of Health and Human Services through contracts HHSN268201100046C, HHSN268201100001C, HHSN268201100002C, HHSN268201100003C, HHSN268201100004C, and HHSN271201100004C. The authors thank the WHI investigators and staff for their dedication, and the study participants for making the program possible. A listing of WHI investigators can be found at: https://www.whi.org/researchers/Documents%20%20Write%20a%20Paper/WHI%20Investigator%20Long%20List.pdf. ARIC: The Atherosclerosis Risk in Communities study has been funded in whole or in part with Federal funds from the National Heart, Lung, and Blood Institute, National Institutes of Health, Department of Health and Human Services (contract numbers HHSN268201700001I, HHSN268201700002I, HHSN268201700003I, HHSN268201700004I and HHSN268201700005I), R01HL087641, R01HL086694; National Human Genome Research Institute contract U01HG004402; and National Institutes of Health contract HHSN268200625226C. The authors thank the staff and participants of the ARIC study for their important contributions. Infrastructure was partly supported by Grant Number UL1RR025005, a component of the National Institutes of Health and NIH Roadmap for Medical Research. CARDIA: The Coronary Artery Risk Development in Young Adults Study (CARDIA) is conducted and supported by the National Heart, Lung, and Blood Institute (NHLBI) in collaboration with the University of Alabama at Birmingham (HHSN268201800005I & HHSN268201800007I), Northwestern University (HHSN268201800003I), University of Minnesota (HHSN268201800006I), and Kaiser Foundation Research Institute (HHSN268201800004I). This manuscript has been reviewed by CARDIA for scientific content. MVP: This research is based on data from the MVP, Office of Research and Development, Veterans Health Administration, and was supported by funding from the Department of Veterans Affairs of Research and Development, Million Veteran Program Grant (1I0101BX003340, 1I01BX003362, and 1I01CX001025) and the NIH (T32 HL007734, K01HL125751, R01HL127564). The content of this manuscript does not represent

including multi-ethnic studies in identification and refinement of lipids-associated loci, which will help to reduce the existing disparities and to pave the road to precision medicine.

## Introduction

Circulating levels of lipids such as high-density lipoprotein cholesterol (HDL-C), low-density lipoprotein cholesterol (LDL-C), total cholesterol (TC), and triglycerides (TG), are associated with atherosclerotic cardiovascular disease, type 2 diabetes, and fatty liver disease [1–3]. Plasma lipid levels are heritable polygenic traits, with twin studies estimating narrow-sense heritability from 0.48 to 0.76 [4]. Genetic association studies have identified over 400 loci associated with lipid traits [5–10,13]. However, the majority of these findings were based on European ancestry populations, and African American (AA), Hispanic, and other minority populations are underrepresented in these studies.

Previous studies have demonstrated distinct lipid profiles in minority populations compared to Europeans, with higher HDL-C and lower TG levels in African ancestry populations and lower levels of HDL-C and TC in Hispanics [11, 12]. In addition, ancestry-specific variants at established lipid loci have been identified in AA and Hispanic ancestry populations, which were monomorphic or with extremely low minor allele frequencies (MAFs) in populations of European descent [13]. The phenotypic variance explained by the established loci is considerably lower in American minority (8.8–12.3%) than in European ancestry populations (12.9–27.8%) [7, 13], demonstrating that discovery and fine-mapping in non-European populations has fallen behind and that focused efforts in these populations are needed.

The Population Architecture using Genomics and Epidemiology (PAGE) Study funded by the National Human Genome Research Institute and the National Institute on Minority Health and Health Disparities was designed to characterize the genetic architecture of complex traits among underrepresented minority populations through large-scale genetic epidemiological research [14]. As part of this initiative, we developed the Multiethnic Genotyping Array (MEGA) to improve fine-mapping and discovery by increasing variant coverage across multiple ethnicities [15]. Using this array in PAGE enabled us to perform genomic analyses to identify lipid loci that may have been missed by previous Euro-centric GWAS, and to explore the generalizability and heterogeneity of the previous findings across major U.S. ethnic groups. We also performed a meta-analysis combining PAGE results with those from the European Network for Genetic and Genomic Epidemiology (ENGAGE) Consortium [16] and other available European GWAS to search for additional novel lipid loci, and sought replication of our new findings in the Million Veteran Program (MVP) [13], the Global Hispanic Lipids Consortium, and the Electronic Medical Records and Genomics (eMERGE) Network [17], the Kaiser Permanente Research Bank [18], the Jackson Heart Study (JHS) [19], and the UK Bio-Bank (UKBB, https://www.ukbiobank.ac.uk/).

## Results

### Study overview

Data on 45,698 participants were included in the minority meta-analysis with 17,641 AAs, 22,830 Hispanics, 2,387 East Asians, 1,912 Native Hawaiians, 604 Native Americans and 333 others (primarily South Asian, mixed heritage, and other racial/ethnic groups) [20]. These participants were drawn from six large well-characterized epidemiological studies: the

the views of the Department of Veterans Affairs or the United States Government. The eMERGE network (Phase I & II): the eMERGE Network was initiated and funded by NHGRI through the following grants: U01HG006828 (Cincinnati Children's Hospital Medical Center/Boston Children's Hospital); U01HG006830 (Children's Hospital of Philadelphia); U01HG006389 (Essentia Institute of Rural Health, Marshfield Clinic Research Foundation and Pennsylvania State University); U01HG006382 (Geisinger Clinic); U01HG006375 (Group Health Cooperative/ University of Washington); U01HG006379 (Mayo Clinic); U01HG006380 (Icahn School of Medicine at Mount Sinai); U01HG006388 (Northwestern University); U01HG006378 (Vanderbilt University Medical Center); and U01HG006385 (Vanderbilt University Medical Center serving as the Coordinating Center) with U01HG004438 (CIDR) and U01HG004424 (the Broad Institute) serving as Genotyping Centers. eMERGE network (Phase III): this phase of the eMERGE Network was initiated and funded by the NHGRI through the following grants: U01HG8657 (Kaiser Washington/University of Washington); U01HG8685 (Brigham and Women's Hospital); U01HG8672 (Vanderbilt University Medical Center); U01HG8666 (Cincinnati Children's Hospital Medical Center); U01HG6379 (Mayo Clinic); U01HG8679 (Geisinger Clinic); U01HG8680 (Columbia University Health Sciences); U01HG8684 (Children's Hospital of Philadelphia); U01HG8673 (Northwestern University); U01HG8701 (Vanderbilt University Medical Center serving as the Coordinating Center); U01HG8676 (Partners Healthcare/Broad Institute); and U01HG8664 (Baylor College of Medicine). JHS: The Jackson Heart Study is supported and conducted in collaboration with Jackson State University (HHSN268201800013I), Tougaloo College (HHSN268201800014I), the Mississippi State Department of Health (HHSN268201800015I) and the University of Mississippi Medical Center (HHSN268201800010I, HHSN268201800011I and HHSN268201800012I) contracts from the National Heart, Lung, and Blood Institute (NHLBI) and the National Institute for Minority Health and Health Disparities (NIMHD). The authors also wish to thank the staffs and participants of the JHS. The views expressed in this manuscript are those of the authors and do not necessarily represent the views of the National Heart, Lung, and Blood Institute; the National Institutes of Health; or the U.S. Department of Health and Human Services. UKBB: This research has been conducted using the UK Biobank Resource (access number: 42680). The funders had no role in study design, data collection and

Atherosclerosis Risk in Communities Study (ARIC), the BioME Biobank (BioMe), the Coronary Artery Risk Development in Young Adults Study (CARDIA), the Hispanic Community Health Study/Study of Latinos (HCHS/SOL), the Multiethnic Cohort Study (MEC), and the Women's Health Initiative (WHI). An additional 22,887 European ancestry participants from ARIC, BioMe, CARDIA, and WHI with individual level data in PAGE were included in the minority plus European meta-analyses along with publicly available summary statistics from ENGAGE in over 62,000 participants (http://diagram-consortium.org/2015_ENGAGE_1KG/) [16]. Across the five major ancestral groups in PAGE, the lowest HDL-C level was observed in Native Hawaiians, the highest LDL-C level was observed in Europeans, and the highest TC and TG levels were observed in Native Americans (Table 1 and S1 Table).

## Identification of novel loci

In the first discovery stage, we performed a minority-centric analysis that included 45,698 non-European ancestry participants in PAGE and conducted a fixed-effect inverse-variance-weighted meta-analysis (S1 Fig). We identified four novel loci for HDL-C (*5q31*-rs17102282, *DLC1*-rs11782435, *ZCCHC6*-rs145312881 and *DDHD1*-rs75405126) and one novel locus for TG (*MTHFD2*-rs182013227) (Table 2 and S2 Fig). These five novel loci remained genome-wide significant after adjustment for previously established variants on the same chromosome ($P_{condition}$<5.0E-8), and none of them exhibited significant evidence of heterogeneity across studies (S2 Table). Ancestry-stratified analysis revealed which ancestral population contributed most to these novel loci (S3 Table). *MTHFD2*-rs182013227 is only polymorphic in AAs (MAF = 0.004). *DDHD1*-rs75405126 is monomorphic in AA and Hispanic populations and was mainly driven by the signal in Native Hawaiians (MAF = 0.008, $P_{Hawaiian}$ = 2.7E-7, S3 Table). In addition to the five novel loci from the minority-combined meta-analysis, we discovered one novel TC locus, *PCSK1*-rs903381, in the Hispanic-specific meta-analysis (Table 2 and S2 Fig).

In the second discovery stage, we performed a minority plus European ancestry meta-analysis in over 131,000 participants (S1 Fig). We identified three additional novel loci (*HLF*-rs12940636 for HDL-C, *B4GALNT3*-rs35882350 for LDL-C and TC, and *GPCPD1*-rs3747910/ rs199986018 for LDL-C/TC), which remained genome-wide significant after adjusting for all established variants ($P_{condition}$<5.0E-8, Table 2 and S4 Table). The lead variants at *GPCPD1* locus are highly correlated ($r^2$ = 0.97). All these loci exhibited evidence of association in both minority and European ancestry populations, showing no significant evidence of heterogeneity across studies (S4 Table) and were common across ancestral groups (Table 2). Of the five novel loci that we identified in the PAGE minority meta-analysis, the association signals were either attenuated (*5q31*, *DLC1* and *DDHD1*) or the variants were not available in European ancestry populations due to low MAF (i.e. *ZCCHC6*) or monomorphism (i.e. *PCSK1* and *MTHFD2*).

Next, we sought replication for our nine newly discovered loci listed in Table 2 in MVP, the Global Hispanic Lipids Consortium, eMERGE, Kaiser, JHS, and UKBB. Details about these replication studies are provided in the S1 Text. For each novel locus, study-specific results were combined through sample-size-weighted meta-analyses. Three novel loci were successfully replicated (*HLF*, *B4GALNT3*, and *GPCPD1*, $P_{discovery+replication}$<5.0E-8), and two additional novel loci exhibited suggestive evidence of replication (*5q31* and *DLC1*, $P_{replication}$<0.05), with smaller effect estimates in the replication results (S5 Table). The failure to replicate the *DDHD1* locus, which was driven by Native Hawaiian signals, may resulted from the absence of ancestry-matched replication studies.

analysis, decision to publish, or preparation of the
manuscript.

**Competing interests:** The authors have declared
that no competing interests exist.

## PrediXcan

We performed a PrediXcan analysis to identify associations between lipid traits and the heritable component of gene expression (GREx) in liver, adipose tissue, and whole blood. GREx of *SCN11A*, which does not map to any known regions, showed significant association with TG in visceral adipose tissue (*P* = 1.4E-6, S6 Table). *SCN11A* gene encodes a member of the voltage-gated sodium channel alpha subunit and is responsible for the generation and propagation of action potentials in neurons and muscles [21]. Replication is needed to confirm this novel finding. In addition, we identified 37 genes mapped to 19 previously reported loci that exhibited significant associations with at least one of the four lipid traits (*P*<2.0E-6, S6 Table). Among the 19 previously reported loci, 16 of them corresponded to expected biological candidate gene(s) while long intergenic noncoding RNAs (*AC067959.1* near *APOB*, and *AP000770.1* and *AP006216.11* near *APOA5*) and pseudogene (*HNRNPA1P10* near *DOCK6/ANGPTL8*) reached significance at the other three loci (S6 Table). Failure to detect associations with well-established candidate genes for some known loci likely reflected, at least in part, attenuation in statistical power as a result of small reference sample sizes or from applying weights derived from primarily European reference transcriptomes to minority populations [22]. Identification of associations with genes other than the candidate genes in these regions likely reflected co-regulation of variant predictors and correlation of gene expressions.

## Evaluation of previously established loci in PAGE minority ancestry populations

We focused the evaluation of 433 previously reported loci in 33,063 minority ancestry participants uniformly genotyped by the MEGA array because of its higher coverage of variants of diverse ancestral groups and better imputation quality [15]. Associations of 276 and 839 unique variants in 30 and 240 previously established loci with at least one lipid trait were identified at *P*≤5.0E-8 and *P*≤0.05 level, respectively (S7 Table). Estimation of allelic heterogeneity of the established loci through inclusion of the SNP by principal component (SNP×PC) interaction term in the main model in MEGA minority populations revealed significant heterogeneity for 31 variants at six replicated loci (*CETP*, *TOMM40*, *TRIB1*, *BUD13*, *GCKR* and *GFOD2*) after Bonferroni correction ($P_{SNP×PC}$≤1.8E-4, 0.05/276 SNPs, S7 Table).

We then evaluated overall trends of the strength of the effect estimates at all known GWAS loci reported by the Global Lipids Genetic Consortium (GLGC) [7] across AA, Hispanic, and European ancestry populations. For all four lipid traits, the effect estimates from Hispanic ancestry populations showed stronger correlation with those from European ancestry populations than the effect estimates from AA ancestry populations (Fig 1). The phenotypic variance explained by these reported variants ranged from 13.1%-23.9%, 12.5%-20.7%, 11.6%-17.0%, and 8.89%-19.6% for HDL-C, LDL-C, TC and TG, respectively, across AA, Hispanic, and European ancestry populations, with the lowest variance explained seen in AA populations (S8 Table).

We explored independent signals in our minority ancestry populations at previously known loci through step-wise conditional analysis adjusting for the most significant hit at each round. We identified independent signals at 12 known loci, and eight of them harbored variants that were monomorphic in European ancestry populations (*ABCA1*, *APOA5*, *CETP*, *LCAT*, *PCSK9*, *LDLR*, *APOE* and *TM6SF2*, S9 Table). Among these eight loci, six of them harbored either missense (*ABCA1*-rs9282541, *APOA5*-rs142953140/rs147210663, *LCAT*-rs35673026, *PCSK9*-rs28362263, *TM6SF2*-rs142056540 and *APOE*-rs769455) or loss of function (LoF) variants (*PCSK9*-rs28362286/rs67608943, S9 Table).

**Table 1. Characteristics of the ancestrally diverse populations in PAGE [1].**

| | African American | Hispanic | East Asian | Native Hawaiian | Native American | Other [3] | European |
|---|---|---|---|---|---|---|---|
| N | 17,641 | 22,830 | 2,378 | 1,912 | 604 | 333 | 22,887 |
| Age (years) | 57.9±12.6 | 53.0±14.9 | 65.6±11.4 | 64.3±8.2 | 61.0±7.9 | 47.0±14.7 | 59.0±12.6 |
| Female (%) | 80.8 | 64.8 | 58.3 | 50.2 | 98.0 | 47.1 | 76.1 |
| HDL (mg/dL) | 54.9±16.2 | 48.6±14.4 | 50.1±17.5 | 41.1±15.4 | 53.0±13.2 | 50.4±18.1 | 51.9±15.0 |
| LDL (mg/dL) | 142.3±43.7 | 132.5±39.6 | 140.9±38.8 | 144.3±36.6 | 139.9±38.7 | 123.3±40.9 | 144.9±39.9 |
| TC (mg/dL) | 218.5±47.5 | 209.9±46.4 | 218.0±42.4 | 210.3±39.3 | 225.0±43.6 | 199.0±45.5 | 224.3±44.8 |
| ln(TG (mg/dL)) [2] | 4.60±0.50 | 4.86±0.55 | 4.79±0.53 | 4.72±0.53 | 4.95±0.53 | 4.86±0.54 | 4.80±0.53 |

[1] Values are shown as mean±SD.

[2] Triglyceride levels were natural-log transformed.

[3] Primarily South Asian, mixed heritage, and other racial/ethnic groups [20].

Next, we performed fine-mapping of previously reported loci leveraging the relatively shorter LD ranges in AA populations and using FINEMAP [23]. In each locus, the number of variants in strong LD ($r^2 \geq 0.6$) with the lead variant in AA participants genotyped on MEGA were calculated using European- and AA-specific LD matrices from the 1000 Genome Phase 3 data. Among the 11 reported loci that showed genome-wide significance with at least one lipid trait in MEGA AA populations, all loci showed a reduction in the number of SNPs except for *APOE* and *LDLR* associated with HDL-C (S10 Table). The most substantial refinement was observed at *APOA5* locus associated with TG, where the number of variants that were highly correlated with the lead variant rs3135506 reduced from 49 (based on European-specific LD) to zero (based on AA-specific LD, S10 Table). This observation is consistent with the findings in our published paper using Metabochip data [24]. We then calculated the 99% credible sets

**Table 2. Novel loci identified in the discovery stage.**

| SNP | chr:pos | Gene | CA/NCA | CAF (%) (AA/HA/EA) | Minority meta-analysis BETA | Minority meta-analysis P | AA-specific meta-analysis BETA | AA-specific meta-analysis P | HA-specific meta-analysis BETA | HA-specific meta-analysis P | Minority + European BETA | Minority + European P | Combined replication [1] Z | Combined replication [1] P | Discovery + Replication Z | Discovery + Replication P |
|---|---|---|---|---|---|---|---|---|---|---|---|---|---|---|---|---|
| HDL | | | | | | | | | | | | | | | | |
| rs17102282 | 5:144103408 | *5q31* | A/G | 47/25/12 | 0.040 | 3.3E-8 | 0.044 | 4.9E-5 | 0.045 | 3.2E-5 | 0.023 | 1.1E-5 | 2.3 | 0.022 | 3.8 | 1.7E-4 |
| rs11782435 | 8:13536115 | *DLC1* | T/C | 8.9/13/22 | 0.063 | 6.8E-10 | 0.014 | 0.47 | 0.085 | 8.8E-10 | 0.023 | 8.6E-6 | 2.6 | 9.6E-3 | 4.1 | 4.6E-5 |
| rs145312881 | 9:89053469 | *ZCCHC6* | A/G | 0.37/0/0 | 0.50 | 1.6E-8 | 0.46 | 1.1E-6 | - | - | - | - | 0.93 | 0.35 | 3.7 | 1.8E-4 |
| rs75405126 [2] | 14:53797383 | *DDHD1* | T/C | 0/0/0.10 | 0.76 | 2.1E-8 | - | - | - | - | 0.49 | 8.8E-5 | -0.16 | 0.88 | 3.2 | 1.2E-3 |
| rs12940636 | 17:53400110 | *HLF* | C/T | 20/24/34 | 0.025 | 1.3E-3 | 0.035 | 9.2E-3 | 0.025 | 0.025 | 0.025 | 9.1E-9 | 4.5 | 6.6E-6 | 6.3 | 2.3E-10 |
| LDL | | | | | | | | | | | | | | | | |
| rs35882350 | 12:623129 | *B4GALNT3* | G/A | 19/16/25 | 0.033 | 1.2E-3 | 0.040 | 9.9E-3 | 0.026 | 0.054 | 0.033 | 1.2E-10 | 6.3 | 4.1E-10 | 8.2 | 3.0E-16 |
| rs3747910 | 20:5528518 | *GPCPD1* | G/A | 25/17/21 | -0.028 | 2.4E-3 | -0.033 | 9.4E-3 | -0.026 | 0.038 | -0.028 | 2.1E-8 | -5.6 | 2.5E-8 | -7.2 | 5.3E-13 |
| TC | | | | | | | | | | | | | | | | |
| rs903381 | 5:95399878 | *PCSK1* | A/G | 3.8/0.85/0 | 0.10 | 2.6E-4 | 0.039 | 0.17 | 0.34 | 2.8E-8 | - | - | -0.78 | 0.43 | 0.44 | 0.66 |
| rs35882350 | 12:623129 | *B4GALNT3* | G/A | 19/16/25 | 0.028 | 3.8E-3 | 0.027 | 0.078 | 0.025 | 0.056 | 0.028 | 2.0E-8 | 5.8 | 8.0E-9 | 7.4 | 9.8E-14 |
| rs199986018 | 20:5544985 | *GPCPD1* | C/A | 24/17/21 | -0.038 | 2.0E-4 | -0.036 | 4.6E-3 | -0.037 | 2.8E-3 | -0.038 | 1.4E-8 | -4.9 | 9.6E-7 | -6.4 | 1.5E-10 |
| TG | | | | | | | | | | | | | | | | |
| rs182013227 | 2:74612959 | *MTHFD2* | C/T | 0.39/0/0 | 0.54 | 5.7E-10 | 0.57 | 3.6E-9 | - | - | - | - | 0.73 | 0.46 | 2.7 | 6.1E-3 |

SNP, single nucleotide polymorphism; chr, chromosome; pos, position; CA, coding allele; NCA, non-coding allele; CAF, coding allele frequency; AA, African American; HA, Hispanic-ancestry; EA, European-ancestry.

[1] The combined replication results were generated through meta-analyzing all replication datasets. Ancestry-specific replication results are shown in S5 Table.

[2] This variant was mainly driven by signal in Hawaiians (MAF = 0.79%, BETA = 0.95, $P_{Hawaiian}$ = 2.7E-7) and details are presented in S3 Table.

for the loci reported by GLGC [6] using FINEMAP. We calculated the credible sets based on GLGC-only data and GLGC plus PAGE minority combined data. After adding PAGE minority participants, 67% of the known loci showed a reduced number of SNPs in the 99% credible sets or reduced length of the 99% credible sets, and 59% of the known loci exhibited over 10% reduction. The most substantially fine-mapped region was the *VLDLR* region, for which the number of SNPs included in the credible set was reduced from over 2000 to one (rs3780181); recently published evidence supports it as the best candidate functional variant at this locus [25]. In addition, the only SNP in the 99% credible sets at *CETP* and *SORT1* loci after including the PAGE minority data was the top hit in the AA- and Hispanic-specific analyses (*CETP*-rs183130 and *SORT1*-rs12740374) rather than the top hit in GLGC (S11 Table). In fine-map analyses of the nine novel loci, 99% credible sets were estimated based on minority meta-analysis results using FINEMAP; the variants included in each credible set are shown in S12 Table.

We further built multiple constructs of weighted polygenic risk scores (PRSs) based on previously reported loci in European ancestry populations for each lipid trait and implemented 10-fold cross validation to evaluate their performance in minority populations. Reported loci from GLGC [6] were used and three PRS constructs were evaluated: (1) PRS1 using reported variants, weighted by the reported effect estimates from GLGC; (2) PRS2 using reported variants from GLGC, weighted by effect estimates observed in PAGE; (3) PRS3 using the most significantly associated variant from PAGE at each reported locus (see Methods section), weighted by PAGE effect estimates. For all four lipid traits, PRS2 explained more variance compared to PRS1 in terms of lower residual values (S13 Table). PRS3 explained more variance of TG compared to PRS2 while no improvement was observed for HDL-C, LDL-C or TC (S13 Table).

## Functional annotation of the novel loci

Bioinformatic follow-up of the novel loci listed in Table 2 was performed using a comprehensive annotation database constructed from whole genome sequence annotator (WGSA) [26] and a custom UCSC analysis hub visualizing enhancer and repressor activities, DNase I hypersensitive sites (DHS) and transcribed regions in adult liver and adipose tissue, which facilitated prioritization of putative functional genes and variants. At the *HLF* locus, the index SNP is located at the 3'UTR region of *HLF* (encodes a member of the proline and acidic-rich protein family, a subset of the bZIP transcription factors) and shows a DANN rank score of 0.90 ($\geq$0.9, possibly deleterious) [27]. *In vitro* experiments demonstrated reduced cellular lipid content after knockdown of *HLF* [28]. Of note, the *TOM1L1* gene is located ~400kb away from the index SNP, which belongs to the same family of the previously reported *TOM1* gene (S3 Fig). At the *B4GALNT3* locus, which was associated with both LDL-C and TC levels, an LD proxy (rs34019521, $r^2 = 0.85$) of the lead variant overlapped with enhancer activity in liver and showed an Eigen PC phred score of 22.11 ($\geq$17, functional) [29]. This variant was part of the 99% credible set of this locus in the fine-map analysis (S12 Table). At each of the nine novel locus, the index SNP and its proxies ($r^2 \geq 0.4$) with a DANN rank score $\geq$0.9 (deleterious), Eigen PC phred score $\geq$17 (functional) [29], eQTL in GTEx database [30], or overlapped with enhancer, repressor, DHS and transcribed regions are summarized in S14 Table. Visualization of novel loci using our custom UCSC analysis hub are presented in S3 Fig.

## Discussion

Employing a two stage meta-analysis, first in our ancestrally diverse minority populations, then combining with populations of European descent, we identified nine novel loci, five of which showed evidence of replication in independent studies. We further identified a novel

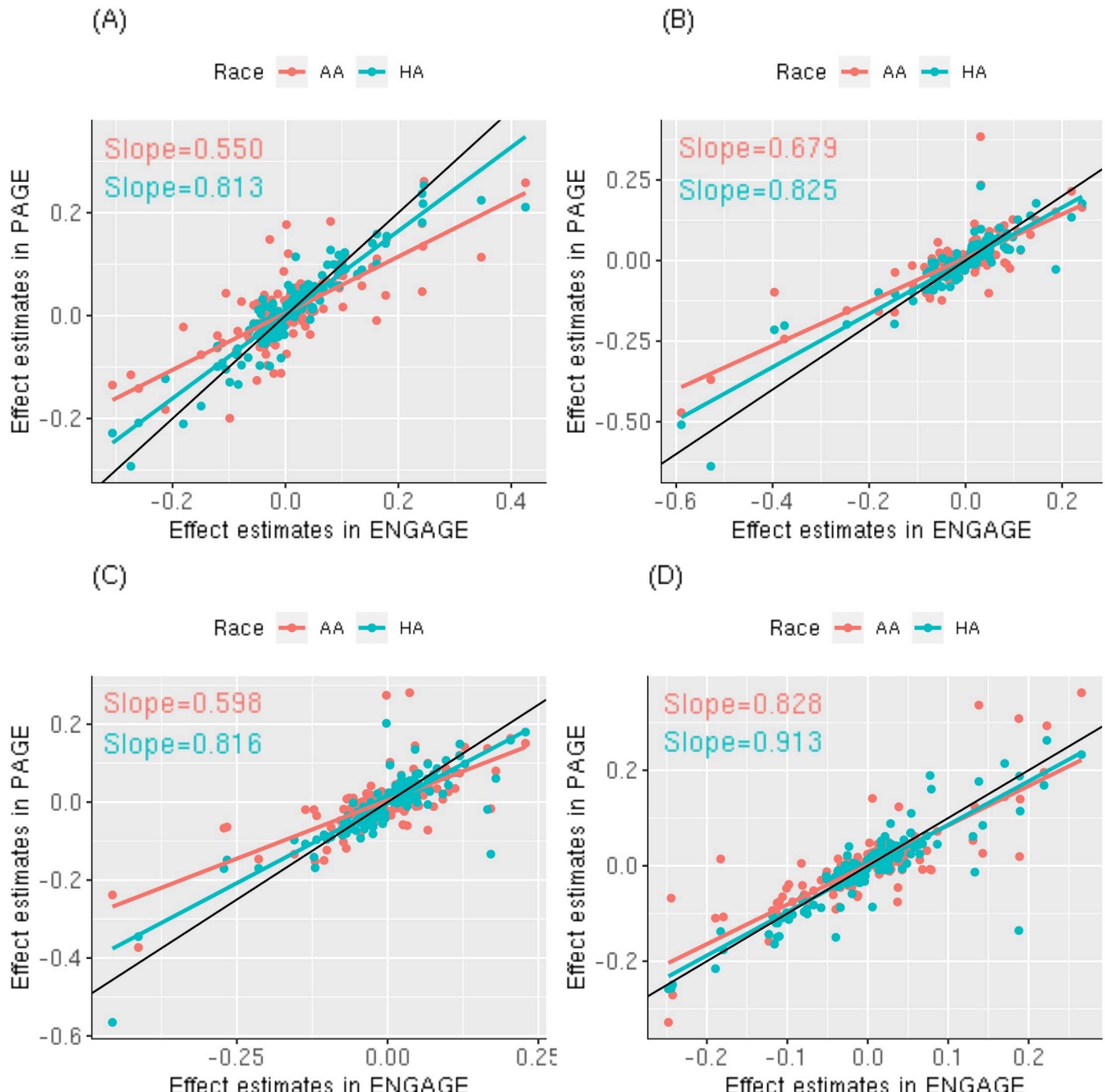

**Fig 1. Comparison of effect estimates across AA, Hispanic and European ancestry populations.** A total of 444 independent variants in 250 loci previously reported by GLGC were included in the comparison. (A) HDL-C; (B) LDL-C; (C) TC; (D) TG.

gene *SCN11A*, and strong candidate target genes in previously known loci using the PrediXcan approach. Independent minority-specific signals at eight previously established loci were identified in the conditional analysis using individual level data. The systematic evaluation of

previously reported loci in PAGE revealed shared genetic background across ethnicities and heterogeneity of allelic effects between minority and European ancestry populations.

Our findings demonstrated three benefits of performing GWAS for lipid traits in ancestrally diverse populations focusing on non-European participants. The first benefit lies in the refinement of previously established loci in terms of more accurate estimation of the effect sizes in non-Europeans. In the examination of reported loci using PRS, we demonstrated that PRSs which were weighted on minority-specific effect sizes explained more variance of all four lipid traits in our non-European populations than the ones weighted on effect sizes reported by GLGC, which focused on participants of European descent. PRSs which used the top hit at each locus from the minority-specific results and weighted on their effect sizes explained even more variance of TG level, although no improvement was observed for the other three lipid traits, which may be the result of insufficient samples sizes in our current analyses. In the comparison of effect estimates for each single variant across ancestral groups, we discovered an attenuation in the magnitude of effects, especially in AA participants, which reflected less European admixture in AA compared to Hispanic populations.

The second benefit is related to the potential of pinpointing the functional variants. Our minority population enabled us to narrow in on a potential functional variant rs3135506 at the *APOA5* locus associated with TG through LD-based fine-mapping. This index SNP is a missense variant (p.Ser19Trp) with a DANN rank score of 0.995 (deleterious), and exhibits suggestive evidence of association with coronary artery disease ($P$ = 7.8E-4) [31]. In a previous exome chip analysis of Norwegian participants with a smaller sample size, this missense variant was no longer significant after accounting for the GWAS index SNP rs964184 [32]. However, we discovered that rs3135506 was much more significant than rs964184 in association with TG ($P$ = 1.00E-23 and 1.5E-4, respectively) and the association result of rs3135506 remained almost unchanged after conditioning on rs964184 ($P$ = 1.0E-23 and 2.6E-20 before and after conditional analysis, respectively) in our AA populations. These results reflected different LD patterns across ancestral groups ($r^2$ = 0.02 and 0.37 in African and European ancestry population from 1000 Genome Phase 3 data, respectively) and emphasized that extra caution would be needed when generalizing results across ancestral populations. Another example is the successful refinement of the *VLDLR* locus, with only one variant rs3780181 in the combined 99% credible set. This variant was recently reported for regulating enhancer activity and *VLDLR* gene expression, supporting it as the best candidate functional variant for this signal [25].

The third benefit was the identification of ancestry-specific variants both at novel and previously reported loci. Among the five novel loci identified in the minority meta-analysis, four of them were either monomorphic (*PCSK1* and *MTHFD2*) or had extremely low MAFs (*DDHD1* and *ZCCHC6*) in European ancestry populations. At the *DDHD1* locus, the novel association was mainly driven by signal from Native Hawaiian ancestry populations that have not been examined before. The minor allele T of rs75405126 is more common in Native Hawaiians than in Europeans (MAF = 0.008 and 0.001, respectively). It also shows fairly high MAF in Asians (0.007), and it is monomorphic in AA and Hispanic ancestry populations. A previous study has shown that expression of *DDHD1* in leukocyte was correlated with plasma HDL-C level [33]. Nevertheless, further investigation and successful replication of these ancestry-specific novel loci are needed. In addition, the availability of individual level data for all minority participants in PAGE enabled us to perform more accurate conditional analyses, leading to identification of minority-specific independent signals at eight previously reported loci. These signals were monomorphic in European ancestry populations, and six of them harbored either missense or LoF variants, which could be potential targets in future pharmaceutical studies (S9

Table). These findings, that may have been missed by the European-focused GWAS, contribute to a more complete picture of the genetic architecture of lipid metabolism.

Despite the heterogeneity and disparity we observed in our analyses, the evidence of the shared genetic architecture of the four lipid traits among different ancestral groups was overwhelming. Among the independent and significant variants reported by GLGC [7], 21%, 18%, 20% and 25% of them reached genome-wide significance in the minority meta-analysis for HDL-C, LDL-C, TC and TG, respectively, with consistent association directions between minority and European ancestry populations. The total sample size of our minority population was only 15% of that in the GLGC discovery stage, and undoubtedly many more loci will surpass the genome-wide significant threshold with an increased sample size. In addition, the two novel loci (*B4GALNT3* and *GPCPD1*) we identified in the minority plus European ancestry meta-analysis, which showed successful replication, were jointly driven by signals in minority and European ancestry populations. *B4GALNT3* (*beta-1,4-N-acetyl-galactosaminyltransferase 3*) and the previously reported *GALNT2* gene both belong to the N-acetylgalactosaminyltransferases family [34]. *GPCPD1* encodes glycerophosphocholine phosphodiesterase 1, and knockdown of this gene resulted in altered lipid metabolites in cells [35].

Although we represented the most ancestrally diverse minority dataset for lipid traits to date, several limitations need to be mentioned. First, the sample sizes for the Native Hawaiian and Native American ancestry populations were limited, which may have obstructed the discovery of novel loci in these two ancestral groups. Second, we were not well-powered to systematically examine the potentially different effects of lipid traits-associated loci between male and female participants in our minority ancestry populations, with previous studies in European ancestry populations reporting a global difference in SNP effects between sexes [9]. Third, we may not be able to capture potential heterogeneity within ancestral groups. For example, the Hispanic participants in the current analysis came from various regions including Central and South America, Puerto Rico, and Mexico, and were grouped together when performing the association testing. The potential heterogeneity may also contribute to the failure of replication of some novel loci even in ancestrally matched samples. Fourth, the associations of the novel loci and independent signals with lipids and related diseases need to be further explored using functional studies to gain a better understanding of the underlying mechanisms. Finally, using the European reference transcriptome may introduced bias in our PrediXcan analysis, emphasizing the need of data collection and better model construction in minority populations.

In summary, we identified nine novel loci, minority-specific independent signals at eight previously established loci, and one novel gene from the PrediXcan analysis, and observed different effect estimates of the associated variants across ancestral groups, which reinforce the need to conduct genetic association studies in participants of diverse ancestral background. The findings in these currently underrepresented populations will provide new insights into the genetics of lipids and associated diseases, thus paving the road to precision medicine.

## Materials and Methods

### Ethics statement

All studies were approved by local Institutional Review Boards and written informed consent was obtained from each participant. The Fred Hutch Institutional Review Board approved the study with protocol number of 8071.

## Study population

The PAGE study incorporated 45,698 minority participants from ARIC, BioMe, CARDIA, MEC, HCHS/SOL, and WHI with available lipid measurements, primarily AAs, Hispanics, Asians, Native Hawaiians, and Native Americans (Table 1). In addition, a total of 22,887 European ancestry participants with lipid measurements from ARIC, BioMe, CARDIA and WHI were available in PAGE. Detailed characteristics of each study are provided in S1 Table, and detailed information for each study is presented in S1 Text.

## Measurement of lipid levels

HDL-C, TC, and TG levels (mg/dL) in fasting blood were measured while LDL-C levels were calculated using the Friedewald Equation. LDL-C levels were not calculated if the corresponding TG levels were greater than 400mg/dL. Lipid levels were further adjusted for medication by adding a constant based on previous publications [24] (S15 Table). If multiple medications were used, only the largest constant was applied. Participants who were pregnant at blood draw or who had fasted less than 8 hours prior to blood draw were excluded from the analysis. TG levels after adjustment for medication were natural-log transformed. Summary statistics of lipid levels in all PAGE participants and the ancestry/study-stratified results are provided in S1 Table.

## Genotyping, imputation and quality control

The PAGE minority populations were genotyped using two different strategies. A total of 33,063 participants with lipids measurements (10,085 AAs, 17,751 Hispanics and 2,378 Asians, 1,912 Native Hawaiians, 604 Native Americans and 333 others) were genotyped using the MEGA array, which was specifically designed to substantially increase variant coverage across multiple ethnic groups [15]. On the MEGA array, 1,705,969 genetic variants were genotyped. Quality control (QC) filters were applied at both the individual sample and the SNP level. At the individual sample level, samples with evidence of sex discrepancy, Mendelian inconsistency, unexpected duplication/non-duplication, poor performance, DNA mixture, identity issue or restricted consent were excluded. At the SNP level, SNPs meeting the following criteria were excluded: (1) failed the Center for Inherited Disease Research (CIDR) technical filters at John Hopkins University; (2) call rate <98%; (3) discordant calls in study duplicates; (4) >1 Mendelian errors in trio and duos; (5) Hardy-Weinberg $P$<1E-4; (6) sex difference in allele frequency ≥0.2 for autosomes/XY; (7) sex difference in heterozygosity >0.3 for autosomes/ XY; (8) positional duplicates. SNPs that passed QC were further imputed to 1000 Genomes Phase 3 data using SHAPEIT2 and IMPUTE (version 2.3.2), resulting in 39,723,562 imputed SNPs with IMPUTE info score no less than 0.4.

A total of 7,556 AA, 5,079 Hispanic and 22,887 European ancestry participants with lipid measurements from ARIC, BioMe, CARDIA, MEC and WHI were previously genotyped using either Affymetrix or Illumina arrays within each individual study (S1 Text). While the QC filters vary slightly by study, similar criteria as those listed above were used including exclusion of: (1) low call rate <90%; (2) discordant calls in study duplicates; (3) >1 Mendelian errors in trio and duos; (4) Hardy-Weinberg $P$<1E-6; (5) sex difference in allele frequency; (7) positional duplicates; (8) ancestry outliers. The genotype data from these studies was imputed to the 1000 Genome Phase 3 panel using IMPUTE (version 2.3.2) in each study separately, and SNPs with info score less than 0.4 were excluded.

## Statistical analyses

The association analysis in the discovery stage was divided into two stages, the minority meta-analysis and the minority plus European meta-analysis. In the minority meta-analysis, we combined the 33,063 participants genotyped on the MEGA array and the additional 7,556 AAs and 5,079 Hispanics genotyped on different arrays through fixed-effect inverse-variance-weighted meta-analysis in METAL [36]. Before the minority meta-analysis, all participants genotyped on the MEGA array were pooled together for association testing, with adjustment for age, sex, study, self-identified ethnicity as a proxy for cultural background, center, household membership, and the first 10 PCs. Lipid levels were inverse-normally transformed by sex. For the additional 7,556 AAs and 5,079 Hispanics, association analyses were performed in each study separately, with adjustment for age, sex and the first 10 PCs. Lipid levels were inverse-normally transformed in each study/genotyping array by sex. All association analyses were performed using SUGEN, which implements a generalized estimating equation (GEE) method and accounts for relatedness [37]. SNPs with effective sample size (effN, effN = $2 \times MAF \times (1-MAF) \times N \times info$, where MAF was the minor allele frequency, N was the sample size and info was the IMPUTE2 info score) less than 30 in MEGA, or less than 5 in the individual studies were excluded before meta-analysis. In the minority plus European ancestry meta-analysis, we combined 45,698 PAGE minority ancestry participants, 22,887 PAGE European ancestry participants, and publicly available association summary statistics from the ENGAGE Consortium through fixed-effect inverse-variance-weighted meta-analysis in METAL, reaching a total sample size of over 131,000. Sample-size-weighted meta-analyses were also performed; the results are shown in S16 Table. SNPs that were only available in one study were excluded after meta-analysis. Conditional analyses adjusting for previously established loci were performed to determine the independency of the novel loci and to explore residual signals as well. The previously established lipids-associated loci list was hand-curated integrating SNPs indexed in the GWAS Catalog (Access date: September 23, 2018) or identified through non-GWAS arrays with covered part of the genome (metabochip or exomechip) [5–7, 9]. Reported SNPs with $P<0.05$ in our meta-analysis on each chromosome were adjusted in the model to achieve an efficient conditional analysis. All conditional analyses in PAGE were performed using individual level data by SUGEN while conditional analysis in ENGAGE was performed using summary statistics by GCTA-COJO [38]. Since all participants in ENGAGE were of European descent, the LD matrix was estimated from 9,345 European ancestry participants from the ARIC study that were available in PAGE. Novel loci were defined as those that fulfilled all of the three criteria: (1) the lead SNP reached genome-wide significance ($P<5.0E-8$) in both the marginal and conditional analysis; (2) the lead SNP was located more than 500kb of any previously established loci; (3) the lead SNP had at least one neighboring SNP (within ±500kb) showing suggestive genome-wide significance ($P<1.0E-5$).

In the replication stage, summary statistics for the nine novel loci were extracted from MVP, the Hispanic GWAS meta-analysis, eMERGE, Kaiser, JHS and UKBB, respectively, in ancestry-stratified and ancestry-combined (if applicable) manners. A proxy variant (rs537734545, $r^2 = 1$) of the *MTHFD2*-rs186278890 was used in MVP. All replication studies used the similar trait transformation strategy as in the discovery studies except for the Kaiser study (HDL was square-root-transformed, TG was log-transformed, and LDL and TC were not transformed) [18], and sample-size-weighted meta-analyses were performed to combine the summary statistics from each study. Details of each replication study are presented in S1 Text.

When examining the potential heterogeneity of previous known loci in PAGE minority ancestry participants, interaction analyses were performed by including SNP×PC terms for all

the first 10 PCs in the models. Models with and without the interaction terms were compared using the *F*-statistic, and an overall SNP×PC interaction *P* value ($P_{SNP \times PC}$) for each known variant was estimated, which indicated whether the additional variance explained by the interaction terms was statistically significant and represented effect modification driven by genetic ancestry [20]. The $P_{SNP \times PC}$ values for all previously reported variants are shown in S7 Table.

The PRSs were constructed by combining the lipid trait-increasing allele counts of the associated variants (reported by GLGC [6]) weighted by the corresponding effect sizes of each allele. In PRS3, the most significant variants in PAGE at each GLGC reported locus (±500kb) were used. Full list of used variants in the PRSs are shown in S17 Table. Ten-fold cross validation was implemented to estimate the trait variance explained by PRS1, PRS2 and PRS3 using four linear models: (1) model 0 which included all covariates in the association analysis (age, sex, study, self-identified ethnicity, center, household membership and the first 10 PCs); (2) model 1 which included PRS1 in additional to all covariates in model 0; (3) model 2 which included PRS2 in addition to all covariates in model 0; (4) model 3 which included PRS3 in addition to all covariates in model 0. The residual values of the four models were used to determine whether there was improvement in estimating lipid levels using different scores.

Explained phenotypic variance for each genetic variant was calculated using the equation below [39]. Explained phenotypic variance $= \frac{2\beta^2 MAF(1-MAF)}{2\beta^2 MAF(1-MAF) + SE^2\, 2N\, MAF(1-MAF)}$

In the fine-map analyses for previously GLGC reported loci [6] using FINEMAP [23], we first generated the posterior probability (PP) of each variant within ±1Mb regions of the top hits in GLGC, PAGE AA and PAGE Hispanic participants. Only variants that were available in both GLGC and PAGE minority data were included in the analysis. The number of causal variants in each region was set to one in FINEMAP. With a single number of casual variants in each region the results of FINEMAP do not depend on the reference LD. We then constructed the 99% credible set for each reported locus in GLGC, PAGE AA and PAGE Hispanic participants, respectively. The combined PP of each variant were calculated by multiplying PP from each ancestral group and rescaling it based on the sum of the PPs for all variants [(PP_i, AA × PP_i, HA × PP_i, GLGC) / sum_i(PP_i, AA × PP_i, HA × PP_i, GLGC), where i refers to the locus, and the AA and HA results are from PAGE]. We used this approach instead of meta-analyzing these three groups because the sample size of European ancestry population from GLGC (N = 188,577) is overwhelming compared to the ones of AA (N = 17,641) and Hispanic (N = 22,830) ancestry populations in PAGE. The meta-analysis results would be driven mainly by GLGC results, which in turn introduces bias in the fine-map analysis. The approach we used took into account both the association *P* value and the sample size of each group. In the fine-map analyses of the nine novel loci, the LD matrix was estimated using the ethnicity that drove the significance of the locus. In particular, LD estimates for *5q31* and *DDHD1* were generated using all minority ancestry participants genotyped on the MEGA array (N = 51,520) and Native Hawaiian participants genotyped on the MEGA array (N = 3,944), respectively. LD estimations of *HLF*, *B4GALNT3* and *GPCPD1* were generated by combining LD information from all minority ancestry participants genotyped by MEGA array (N = 51,520) and all available European participants from ARIC in PAGE (N = 9,345) through a sample-size-weighted approach. For *DLC1* and *PCSK1* loci, which were driven by Hispanic signals, and *ZCCHC6* and *MTHFD2* loci, which were driven by AA signals, LD estimations were generated using all Hispanic (N = 22,244) and AA (N = 17,324) participants genotyped on the MEGA array, respectively.

### Bioinformatic functional follow-up and PrediXcan analysis

Bioinformatic functional follow-up was performed for each novel locus using our comprehensive functional annotation database and a custom UCSC analysis data hub. In the PrediXcan analysis, we focused on adipose tissue, liver and whole blood, which are closely linked to lipid metabolism. There were 8,271, 6,594, 3,355 and 6,298 genes included in the models in subcutaneous adipose tissue, visceral adipose tissue, liver and whole blood, respectively, and genes with $P<$2.04E-6 [0.05/(8271+6594+3355+6298)] were considered as significant. Details are presented in S1 Text.

## Supporting information

**S1 Fig. QQ plots of the meta-analyses.** (A) HDL for the minority meta-analysis; (B) LDL for the minority meta-analysis; (C) TC for the minority meta-analysis; (D) TG for the minority meta-analysis; (E) TC for the Hispanic-specific meta-analysis; (F) HDL for the minority plus European meta-analysis; (G) LDL for the minority plus European meta-analysis; (H) TC for the minority plus European meta-analysis; (I) TG for the minority plus European meta-analysis.
(DOCX)

**S2 Fig. Locuszoom plots for the nine novel loci.** Genetic coordinates are displayed along the x-axis (hg19) and genome-wide association significance level is plotted against the y-axis as -log10(P value). LD is indicated by color scale in relationship to the most significant SNP (colored as purple diamond) in each association (red: r2≥0.8, orange: 0.6≤r2<0.8, green: 0.4≤r2<0.6, blue: 0.2≤r2<0.4, navy: r2<0.2). (A) 5q31 for HDL; (B) DLC1 for HDL; (C) ZCCHC6 for HDL; (D) DDHD1 for HDL; (E) HLF for HDL; (F) B4GALNT3 for LDL; (G) GPCPD1 for LDL; (H) PCSK1 for TC; (I) B4GALNT3 for TC; (J) GPCPD1 for TC; (K) MTHFD2 for TG.
(DOCX)

**S3 Fig. Functional annotation of the nine novel loci.** The top hit at each locus was colored purple, and the SNPs showing r2≥0.8, 0.8>r2≥0.6, 0.6>r2≥0.4 and 0.4>r2≥0.2 were colored red, orange, green and blue, respectively. (A) 5q31; (B) DLC1; (C) ZCCHC6; (D) DDHD1; (E) HLF; (F) B4GALNT3; (G) GPCPD1; (H) PCSK1; (I) MTHFD2.
(DOCX)

**S1 Table. Characteristics of study samples in PAGE MEGA and non-MEGA studies.**
(XLSX)

**S2 Table. Study-specific and conditional results of the novel loci identified in the minority meta-analysis.**
(XLSX)

**S3 Table. Ethnic-specific results of the novel loci identified in the minority meta-analysis.**
(XLSX)

**S4 Table. Study-specific and conditional results of the novel loci identified in the minority plus European meta-analysis.**
(XLSX)

**S5 Table. Replication of the nine novel loci.**
(XLSX)

**S6 Table. Significant gene-lipid associations identified in the PrediXcan analysis.**
(XLSX)

**S7 Table. Results of previously reported variants in PAGE minorities genotyped on MEGA.**
(XLSX)

**S8 Table. Phenotypic variance explained by the GLGC reported variants.**
(XLSX)

**S9 Table. Independent signals at previously established loci.**
(XLSX)

**S10 Table. Fine-map of reported loci using AA-specific LD.**
(XLSX)

**S11 Table. Fine-map of the GLGC reported loci using FINEMAP.**
(XLSX)

**S12 Table. Fine-map of the nine novel loci using FINEMAP.**
(XLSX)

**S13 Table. Performance of the PRSs for each lipid trait.**
(XLSX)

**S14 Table. Novel loci annotation summary.**
(XLSX)

**S15 Table. Medication adjustment of lipid levels.**
(XLSX)

**S16 Table. Sample-size-weighted meta-analysis results for the nine novel loci.**
(XLSX)

**S17 Table. SNPs used in building the 3 constructs of PRSs.**
(XLSX)

**S1 Text. Supplementary methods.**
(DOCX)

## Acknowledgments

The PAGE consortium thanks the staff and participants of all PAGE studies for their important contributions. We thank Rasheeda Williams and Margaret Ginoza for providing assistance with program coordination. The complete list of PAGE members can be found at http://www.pagestudy.org. Assistance with data management, data integration, data dissemination, genotype imputation, ancestry deconvolution, population genetics, analysis pipelines, and general study coordination was provided by the PAGE Coordinating Center. Genotyping services were provided by the Center for Inherited Disease Research (CIDR). Genotype data quality control and quality assurance services were provided by the Genetic Analysis Center in the Biostatistics Department of the University of Washington.

BioMe: Data of BioMe Biobank used in this study was provided by the Charles Bronfman Institute for Personalized Medicine at the Icahn School of Medicine at Mount Sinai. Phenotype data collection was supported by the Andrea and Charles Bronfman Philanthropies.

National Institute of Neurological Disorders and Stroke, NIH Institution-Office of Dietary Supplements.

WHI: A listing of WHI investigators can be found at: https://www.whi.org/researchers/Documents%20%20Write%20a%20Paper/WHI%20Investigator%20Long%20List.pdf.

ARIC: The authors thank the staff and participants of the ARIC study for their important contributions.

MVP: The content of this manuscript does not represent the views of the Department of Veterans Affairs or the United States Government.

JHS: The authors wish to thank the staffs and participants of the JHS. The views expressed in this manuscript are those of the authors and do not necessarily represent the views of the National Heart, Lung, and Blood Institute; the National Institutes of Health; or the U.S. Department of Health and Human Services.

UKBB: This research has been conducted using the UK Biobank Resource (access number: 42680).

## Author Contributions

**Conceptualization:** Lisa W. Martin, Christopher S. Carlson, Lucia A. Hindorff, Kari E. North, Charles Kooperberg, Iona Cheng, Ulrike Peters.

**Data curation:** Yao Hu, Mariaelisa Graff, Jeffrey Haessler, Steven Buyske, Stephanie A. Bien, Heather M. Highland, Katherine K. Nishimura, Niha Zubair, Yingchang Lu, Marie Verbanck, Lisa W. Martin, Martha Daviglus, Ron Do, Ruth J. F. Loos, Lynne R. Wilkens, Loic Le Marchand, Chris Haiman, Daniel O. Stram, Kari E. North, Charles Kooperberg, Iona Cheng.

**Formal analysis:** Yao Hu, Mariaelisa Graff, Jeffrey Haessler, Steven Buyske, Stephanie A. Bien, Heather M. Highland, Katherine K. Nishimura.

**Funding acquisition:** Tara C. Matise, Christopher S. Carlson, Ruth J. F. Loos, Lynne R. Wilkens, Loic Le Marchand, Chris Haiman, Daniel O. Stram, Kari E. North, Charles Kooperberg, Iona Cheng, Ulrike Peters.

**Methodology:** Yao Hu, Mariaelisa Graff, Jeffrey Haessler, Ran Tao, Jose Luis Ambite.

**Project administration:** Steven Buyske, Tara C. Matise, Lucia A. Hindorff.

**Resources:** Jose Luis Ambite.

**Software:** Ran Tao, Jose Luis Ambite.

**Supervision:** Christopher S. Carlson, Ron Do, Ruth J. F. Loos, Lynne R. Wilkens, Loic Le Marchand, Chris Haiman, Daniel O. Stram, Lucia A. Hindorff, Kari E. North, Charles Kooperberg, Iona Cheng, Ulrike Peters.

**Validation:** Austin T. Hilliard, Derek Klarin, Scott M. Damrauer, Yuk-Lam Ho, Peter W. F. Wilson, Kyong-Mi Chang, Philip S. Tsao, Kelly Cho, Christopher J. O'Donnell, Themistocles L. Assimes, Lauren E. Petty, Jennifer E. Below, Ozan Dikilitas, Daniel J. Schaid, Matthew L. Kosel, Iftikhar J. Kullo, Laura J. Rasmussen-Torvik, Gail P. Jarvik, Qiping Feng, Wei-Qi Wei, Eric B. Larson, Frank D. Mentch, Berta Almoguera, Patrick M. Sleiman, Laura M. Raffield, Adolfo Correa.

**Writing – original draft:** Yao Hu.

**Writing – review & editing:** Mariaelisa Graff, Jeffrey Haessler, Steven Buyske, Stephanie A. Bien, Ran Tao, Heather M. Highland, Katherine K. Nishimura, Niha Zubair, Yingchang Lu,

Marie Verbanck, Austin T. Hilliard, Derek Klarin, Scott M. Damrauer, Yuk-Lam Ho, Peter W. F. Wilson, Kyong-Mi Chang, Philip S. Tsao, Kelly Cho, Christopher J. O'Donnell, Themistocles L. Assimes, Lauren E. Petty, Jennifer E. Below, Ozan Dikilitas, Daniel J. Schaid, Matthew L. Kosel, Iftikhar J. Kullo, Laura J. Rasmussen-Torvik, Gail P. Jarvik, Qiping Feng, Wei-Qi Wei, Eric B. Larson, Frank D. Mentch, Berta Almoguera, Patrick M. Sleiman, Lisa W. Martin, Martha Daviglus, Tara C. Matise, Jose Luis Ambite, Christopher S. Carlson, Ron Do, Ruth J. F. Loos, Lynne R. Wilkens, Loic Le Marchand, Chris Haiman, Daniel O. Stram, Lucia A. Hindorff, Kari E. North, Charles Kooperberg, Iona Cheng, Ulrike Peters.

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
