## [Decision Letter · Decision Letter 0]

8 Jul 2019

Dear Dr Hu,

Thank you very much for submitting your Research Article entitled 'Minority-centric meta-analyses of blood lipid levels identify novel loci in the Population Architecture using Genomics and Epidemiology (PAGE) Study' to PLOS Genetics.

The manuscript was fully evaluated at the editorial level and by an independent peer reviewer. In general, PLOS Genetics seeks input from 3 or more peer reviewers, but we have been unable to secure additional reviewer comments despite many attempts to do so. The reviewer's comments have been assessed and considered by members of the editorial board; we agree with and fully support those comments, and have therefore decided to proceed with the single review in hand.

As you will see, the reviewer (and the editors) appreciate the attention to an important problem, but note that there are some substantial concerns about the current manuscript. We will not be able to accept this version of the manuscript, but we would be willing to review again a much-revised version. Many of the concerns will be straightforward to address, but please note that the concern regarding replication (point 2) is especially important and will need to be fully addressed for a revised manuscript to move forward.

If you decide to revise the manuscript for further consideration at PLOS Genetics, please aim to resubmit within the next 60 days, unless it will take extra time to address the concerns of the reviewers, in which case we would appreciate an expected resubmission date by email to plosgenetics@plos.org.

[LINK]

We are sorry that we cannot be more positive about your manuscript at this stage. Please do not hesitate to contact us if you have any concerns or questions.

Yours sincerely,

Gregory Barsh

Editor-in-Chief

PLOS Genetics

Gregory Copenhaver

Editor-in-Chief

PLOS Genetics

Reviewer's Responses to Questions

**Comments to the Authors:**

Reviewer #1: Hu et al. present results of GWAS analyses of lipid profiles in ethnic minorities in the PAGE Consortium. They identified novel loci, some of which are driven by one ancestry, and/or are rare or monomorphic in Europeans. Some of the novel loci were replicated in additional published ethnic-specific and trans-ethnic studies. Allelic effects on lipids were not heterogeneous across ethnicities. Loci were interrogated for potential causal genes and fine-mapped to localise potential causal variants. The study is very timely, and the focus on the genetics of lipids in minority groups is extremely important. The messages about fine-mapping and use of multi-ethnic risk scores are very important to the human genetics research community. However, I have some concerns over the analytical approaches used.

1. Trait transformations. The authors state that lipid levels were inverse-normally transformed by sex. It was not clear to me whether this meant inverse-rank normalisation, or a Z-score transformation. In either case, this means that the effect estimates obtained for each study are NOT on the same scale, so it was not clear to me that a fixed-effects meta-analysis with inverse-variance weighting of effect sizes was appropriate, and that a Stouffer fixed-effects meta-analysis would be more appropriate. There are also no details (that I could find) on trait transformations in the replication sets, but again the same concern remains about effect size differences between studies.

2. Lack of replication. For several of the novel loci, there is very weak (if any) evidence of replication, even in ethnically matched components. I couldn’t find the sample sizes of the replication sets anywhere in the manuscript, but I’m assuming that they are substantially larger than PAGE? What is the explanation for this lack of replication – without this, I find it difficult to accept these associations as real.

3. Conditional analyses. These were performed using individual level data in PAGE, and approximate conditioning with GCTA in ENGAGE, but what was the reference used for LD.

4. The authors mention testing for heterogeneity in effects due to ethnicity by testing for interaction with PCs in PAGE, but I couldn’t find any information in the methods about this, nor any plots of PCs to demonstrate that this was an appropriate analysis/interpretation. Overall, I found the methods very sparse, and much more detail is needed to allow others to repeat experiments.

5. FINEMAP analyses. My understanding from reading the methods is that FINEMAP with first applied within each ethnic group, and then the results combined across ethnic groups. However, there are no details provided for the LD reference used for these analyses, and the parameter settings used, such as number of causal variants in a region. It is also not clear to me that combining the posterior probabilities across ethnicities as described is appropriate, so some justification of this approach is needed.

6. MTHFD2 locus. The authors highlight a SNP in strong LD with the index SNP, but it si not clear in which population group this LD is calculated? It would be more sensible to quote the relative posterior probability from the fine-mapping analysis, compared to the lead SNP – is it even in the 99% credible set? Similarly, I found listing SNPs that happened to have some interesting annotation that have r2>0.2 with an index SNP unimportant – again, it is not clear in which population the LD has been measured, or the relative posterior probabilities.

7. Table 2. It would be useful to have: MAF for each ethnic-specific analysis; the final combined meta-analysis p-value across discovery and replication; p-values to two significant figures throughout.

**Have all data underlying the figures and results presented in the manuscript been provided?**

Reviewer #1: Yes

PLOS authors have the option to publish the peer review history of their article (what does this mean?). If published, this will include your full peer review and any attached files.

Reviewer #1: No

---

## [Decision Letter · Decision Letter 1]

23 Jan 2020

Dear Dr Hu,

Thank you very much for submitting your Research Article entitled 'Minority-centric meta-analyses of blood lipid levels identify novel loci in the Population Architecture using Genomics and Epidemiology (PAGE) Study' to PLOS Genetics. Your manuscript was fully evaluated at the editorial level and by independent peer reviewers.

The revised manuscript was seen by the original reviewer. As you will see, their comments are generally positive, but there are some remaining concerns that we ask you address in a hopefully final round of minor revision.

We therefore ask you to modify the manuscript according to the review recommendations before we can consider your manuscript for acceptance. Your revisions should address the specific points made by each reviewer.

[LINK]

Yours sincerely,

Gregory Barsh

Editor-in-Chief

PLOS Genetics

Gregory Copenhaver

Editor-in-Chief

PLOS Genetics

Reviewer's Responses to Questions

**Comments to the Authors:**

Reviewer #1: The authors have addressed most of my comments, and I believe the manuscript has been improved. However, I have a couple of outstanding comments...

1. I had not realised that the FINEMAP analysis had been performed assuming a single causal variant at each locus. Given this assumption, it is not clear why the authors have run FINEMAP, since this has been designed to allow for multiple causal variants at the locus. With a single causal variants, I believe that the LD reference is irrelevant, since no joint models with multiple SNPs are fitted. In theory, the association summary statistics for each ethnicity could just be used to calculate a Bayes' factor and posterior probability for each SNP. I think the methods should also state that the posterior probability calculation across ethnicities assumes that there is a single causal variant at the locus, and that this variant is the same in each ethnic group.

2. Heterogeneity assessment using SNP x PC interaction effect. I like this way of assessing heterogeneity due to ancestry, but wasn't clear why 10 PCs were used. Please provide justification. Including 10 PCs (if they are not needed) will substantially reduce power to detect heterogeneity (since an extra degree of freedom is needed for each PC in the test).

3. Tables still do not have p-values presented to two significant figures - Table 2 has a mixture of two and three significant figures (e.g. 4.89E-5 is three significant figures, 0.47 is two significant figures). I think all tables (including supplementary could do with proof reading) - e.g. in ST16, the causal allele frequency of rs11782435 is 2022%?

**Have all data underlying the figures and results presented in the manuscript been provided?**

Reviewer #1: Yes

PLOS authors have the option to publish the peer review history of their article (what does this mean?). If published, this will include your full peer review and any attached files.

Reviewer #1: No

---

## [Editor Report · Decision Letter 2]

19 Feb 2020

Dear Dr Hu,

We are pleased to inform you that your manuscript entitled "Minority-centric meta-analyses of blood lipid levels identify novel loci in the Population Architecture using Genomics and Epidemiology (PAGE) Study" has been editorially accepted for publication in PLOS Genetics. Congratulations!

Yours sincerely,

Gregory S. Barsh

Editor-in-Chief

PLOS Genetics

Gregory Copenhaver

Editor-in-Chief

PLOS Genetics

Comments from the reviewers (if applicable):

**Data Deposition**

http://datadryad.org/submit?journalID=pgenetics&manu=PGENETICS-D-19-00861R2

**Press Queries**

---

## [Editor Report · Acceptance letter]

11 Mar 2020

PGENETICS-D-19-00861R2 

Minority-centric meta-analyses of blood lipid levels identify novel loci in the Population Architecture using Genomics and Epidemiology (PAGE) Study 

Dear Dr Hu, 

We are pleased to inform you that your manuscript entitled "Minority-centric meta-analyses of blood lipid levels identify novel loci in the Population Architecture using Genomics and Epidemiology (PAGE) Study" has been formally accepted for publication in PLOS Genetics! Your manuscript is now with our production department and you will be notified of the publication date in due course.

With kind regards,

Matt Lyles

PLOS Genetics

On behalf of:
